# Paleo-proteomic analysis of Iron Age dental calculus provides direct evidence of Scythian reliance on ruminant dairy

Jaruschka Pecnik[1,2]*, Alicia R. Ventresca Miller[3,4,5], Christian Panse[6,7], Laura Kunz[6], Antje Dittmann[6], James A. Johnson[8], Sergey Makhortykh[9], Ludmilla Litvinova[9], Svetlana Andrukh[10], Gennady Toschev[10], Michael Krützen[11], Verena J. Schuenemann[1,2,11,12,13], Shevan Wilkin[1,2,5,14]*

1 Institute of Evolutionary Medicine, University of Zurich, Zurich, Switzerland, 2 Department of Environmental Sciences, University of Basel, Basel, Switzerland, 3 Department of Anthropology, University of Michigan, Ann Arbor, Michigan, United States of America, 4 University of Michigan Museum of Anthropological Archaeology, University of Michigan, Ann Arbor, Michigan, United States of America, 5 Department of Archaeology, Max Planck Institute for Geoanthropology, Jena, Germany, 6 Functional Genomics Centre Zurich (FGCZ), ETH and University of Zurich, Zurich, Switzerland, 7 Swiss Institute of Bioinformatics (SIB), University of Lausanne, Lausanne, Switzerland, 8 Department of Anthropology, University of Wyoming, Laramie, Wyoming, United States of America, 9 Institute of Archaeology of the National Academy of Sciences Ukraine, Kyiv, Ukraine, 10 Zaporizhzhya National University, Zaporizhzhya, Ukraine, 11 Department of Evolutionary Anthropology, University of Zurich, Zurich, Switzerland, 12 Department of Evolutionary Anthropology, University of Vienna, Vienna, Austria, 13 Human Evolution and Archaeological Sciences (HEAS), University of Vienna, Vienna, Austria, 14 Australian Research Centre for Human Evolution, Griffith University, Brisbane, Queensland, Australia

* jaruschka.pecnik@uzh.ch (JP); shevan.wilkin@iem.uzh.ch (SW)

## Abstract

The Scythians, often described as mounted horse-back warriors of the Iron Age steppe with lavish burial goods, have attracted increasing scientific interest over the past years. Recent genetic and multi-isotopic studies have uncovered that the 'Scythians' were neither a homogenous political nor a cultural group, but rather diverse populations of heterogeneous origins with intricate socio-political systems. Although populational differences in agro-pastoral subsistence regimes of Northern Black Sea Region groups have previously been identified through stable isotope analysis, it remains unclear which animal products were consumed. Here we investigate the dietary systems of two Scythian-era populations in present-day Ukraine using protein analysis of ancient dental calculus. Various dietary proteins and their taxonomic origin were identified revealing the consumption of milk from ruminant and equine species. This study supplements previous findings that Scythians engaged in complex, agro-pastoralist subsistence strategies in forest-steppe and steppe environments.

---

which permits unrestricted use, distribution, and reproduction in any medium, provided the original author and source are credited.

**Data availability statement:** All relevant data are within the manuscript and its Supporting Information files except for mass spectrometry proteomics data, which is publicly accessible on MassIVE (massive.ucsd.edu; MSV000092635).

**Funding:** The author(s) received no specific funding for this work.

**Competing interests:** The authors have declared that no competing interests exist.

## Introduction

During the Iron Age (ca. 700–200 cal. BCE), Scythian warriors occupied vast expanses of the Eurasian steppe, spanning from modern-day Hungary to the Altai mountains [1]. At multiple steppe sites, an ubiquitous prevalence of unique animal-style art (ASA) has been recovered, indicating unprecedented inter-regional interactions between Scythians [2]. However, detailed studies on ancient DNA, multi-isotopic analyses of Scythian populations, and well-established archaeological data have demonstrated that the notion of a Scythian "empire" [3] was actually composed of several heterogeneous groups of multiregional origins with differences in lifestyles, social stratifications, and subsistence strategies [4–11]. Hence, the shared motifs and artistic renditions of ASA that previously suggested nomadic forms of mobility [2], have been reconsidered as indications of regional separations [12]. Within these culturally and genetically distinct populations, Scythian-era groups likely had complex and varied subsistence systems and mobility strategies (ranging from highly mobile to sedentary), with hierarchical societies that included royals, farmers, and artisans. Some of these groups also interacted with neighbouring and distant populations, such as the Greeks, which is indicated by variety of cultural material findings such as pottery or gold objects in Scythian mounded burials (kurgans) and historical records [1,13,14].

The Eurasian steppe encompasses vast expanses of grasslands and is characterised by a continental climate, marked by hot summers and cold winters [15]. Across the steppe, different soil conditions and water resources built the basis of various forms of subsistence strategies and/ or combinations of hunter-gatherer, pastoral and agricultural lifestyles. For pastoralists, the vast grasslands offer ample sustenance for large herds resulting in access to primary (e.g., meat) and secondary animal products (e.g., milk) from livestock such as cattle, goats, sheep, and horses [14]. Since the Early Bronze Age (~ 3,000 BCE), fresh milk has provided a vital hydration resource on the arid environment and enabled steppe populations the conversion of inedible grasses into consumable milk products such as yoghurts and cheeses [16–18]. Apart from the vast grasslands, the Eurasian steppe also contains regions (such as parts of present-day Ukraine and Kazakhstan) where crop cultivation occurs in areas with adequate precipitation or where irrigation is possible. Hence, in such regions, favourable climatic and environmental conditions like nutrient-rich soils [19] and easily accessible water sources [20] enabled some western steppe populations to practise horticulture or agriculture of crops such as wheat, barley, and millet [7]. Although most populations of the Scythian-era engaged in some form of agro-pastoralism supplemented by hunting and gathering [7,21], the particular species exploited for their food systems, especially milk and dairy products remain unknown, and may have varied between populations on the steppe and forest-steppe.

In present-day Ukraine, the sites Bilsk (ca. 700–200 cal. BCE) and Mamai-Gora (ca. 500–200 cal. BCE) (Fig 1) have been identified through historical records, (bio)archaeological findings, and multi-isotope analyses as two important political and economic centres of the Scythian "empire", where inhabitants engaged in agro-pastoralism [7,21–23]. In northeastern Ukraine, Bilsk is located within the

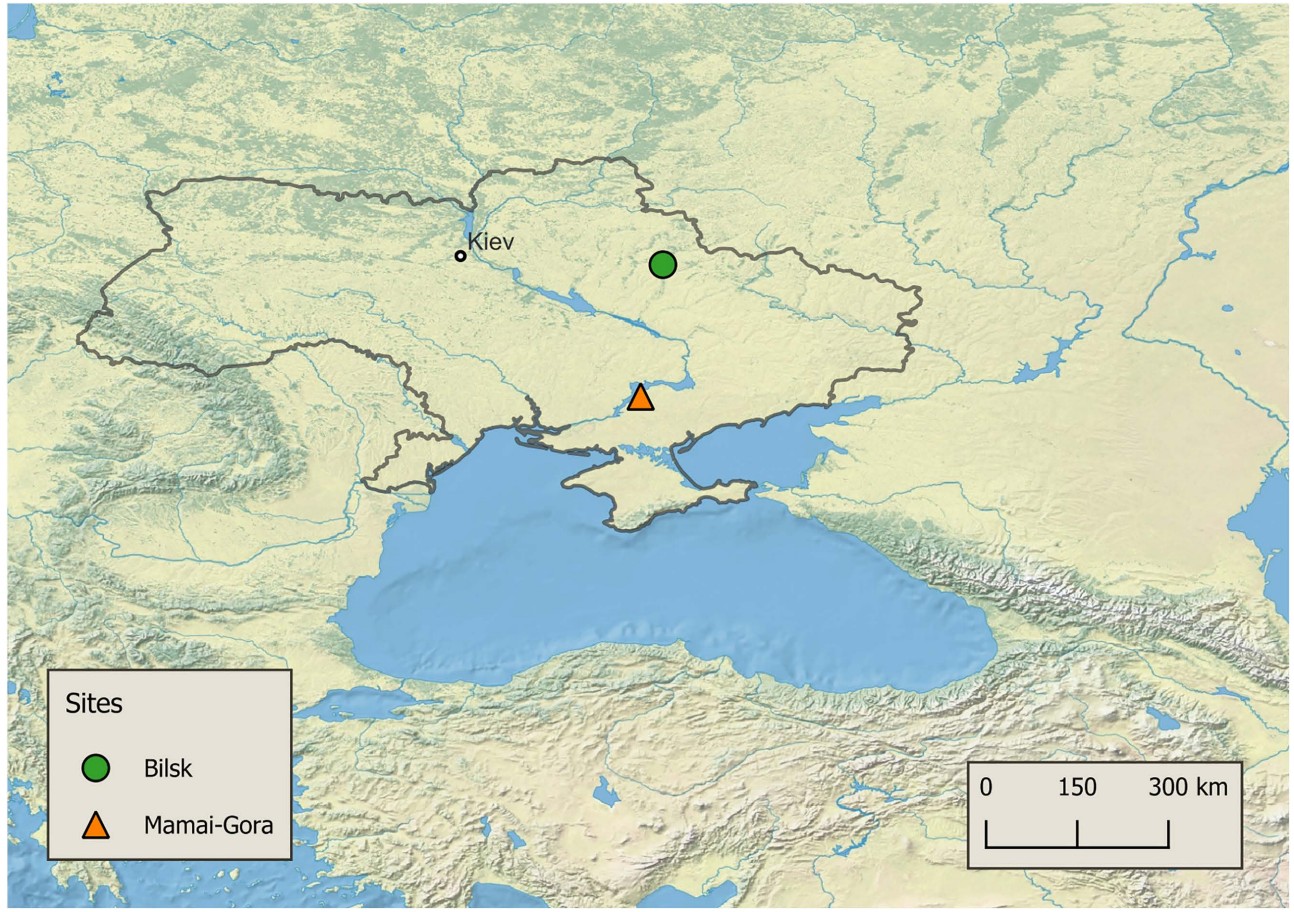

**Fig 1. World map with the location of the Scythian-era sites in the northern Black Sea region in present-day Ukraine.** The map was created using QGIS 3.42.3 [https://qgis.org/en/site/], the free vector and raster map data from Natural Earth [https://www.naturalearthdata.com/], and the coordinates from Google Maps [maps.google.com].

border of forest-steppe and steppe environment and encompasses a huge area of approximately 5000 ha with channels and streams. Over 1000 burials were identified within (e.g., Cemetery B and Tsarina) and outside (e.g., Osnyagi 3[rd] Field, Pereschepino, and Marchenki 8[th] and 9[th] Fields) of the eastern and western fortification walls of Bilsk [19,24,25]. Archaeological findings and historical records indicate that this population consisted not only of a complex stratified social structure, but also operated as an important centre for political authority [19,26]. Several researchers link this site to the historical city Gelonus mentioned by Herodotus (490/480–424 BCE) and associate it with one of the Scythian kings' centre of power [10,19,23,26,27]. Compared with Bilsk, Mamai-Gora is located in south-central Ukraine steppe environment on the left bank of the river Dnieper and has a rather dense necropolis with over 390 explored burials. The majority of the graves have been dated by archaeological material analyses and radiocarbon dating to between 500 and 300 BCE, whereas a few graves were testified to around 700–600 BCE [7,21,22,26,28].

Despite a few stable isotopic analyses on Scythian-era communities that provided broad insights into dietary variation [7,21,29], there has been a marked lack of biomolecular studies that illuminate their dietary practices. Protein analysis from archaeological materials such as ceramic and metal vessels [30–32] or bioarchaeological remains like dental calculus (mineralized dental plaque) [17,33–36] can offer more detailed information about the species specific content of diets, recovered dietary

proteins, and their taxonomic origins through the identification of amino acid sequences [37–39]. Dental calculus is particularly well-suited for analysing past human diets, as its formation rapidly traps proteins from both the host and the surrounding environment, along with other biomolecules, micro-remains, and debris. This layered accumulation on the supra- and subgingival surfaces of teeth throughout an individual's life provides valuable insights into subsistence strategies at both individual and populational level, revealing whether consumed foods were locally sourced or obtained through trade [17,18,34–36,40,41].

To resolve more details about Scythian-era food systems, we report the first proteomic study of dental calculus of 28 individuals from the Scythian-era sites of Bilsk and Mamai-Gora located on in present-day Ukraine (Fig 1). Our results demonstrate that individuals at both sites consumed milk from ruminant herd animals, such as cattle, sheep, and goats. Additionally, horse milk consumption was identified in a single individual at Bilsk at the subsite Tsarina. Based on proteomic evidence of milk and dairy consumption, combined with established stable isotopic data of domestic grain production, we propose that these communities had diverse diets.

## Results

We analysed 43 dental calculus samples (Bilsk: n = 21, Mamai-Gora: n = 22) from 28 individuals (Bilsk: n = 16, Mamai-Gora: n = 12, Table 1) excavated from six different Iron Age subsites at Bilsk (Cemetery B, Marchenki 8th and 9th Fields,

**Table 1. Information about the recovered individuals from Bilsk and Mamai-Gora along with burial context and identified peptides extracted from dental calculus.**

| Individual | Excavation context, bioarchaeological information, and mobility identification | | | | | Dietary protein analysis | | | |
|---|---|---|---|---|---|---|---|---|---|
| | Site | Subsite | Sex | Age estimation | Locality of individual (Sr)* | Identified proteins | PSMs count | Unique peptides | Taxa identified |
| 135 | Bilsk | Cemetery B | M | ~35 y | _ | – | 0 | 0 | – |
| 143 | Bilsk | Tsarina | M | 40 + y | Local[a] | BLG1, BLG, Alpha-S1-casein | 18 | 11 | *Bovidae, Bovinae, Bovinae/Ovis, Caprinae, Ovis, Equus,* unspecific |
| 147 | Bilsk | Cemetery B | M | 30-35y | Local[a] | – | 0 | 0 | – |
| 151-152 | Bilsk | Marchenki 8th Field | M | 45 + y | Local[a] | Alpha-S1-casein | 2 | 2 | *Bovidae, Bovinae* |
| 158 | Bilsk | Tsarina | M | 24-35y | Born in Bilsk, moved away as adolescent and returned as an adult[a] | – | 0 | 0 | – |
| 181-182 | Bilsk | Marchenki 9th Field | F | 45 + y | Born outside of region and moved to Bilsk as a child[a] | BLG, Alpha-S1-casein | 7 | 6 | Pecora, *Bovidae, Bovinae, Bovinae/Ovis* |
| 185-186 | Mamai-Gora | – | M | 20-35y | Local[b] | – | 0 | 0 | – |
| 192 | Mamai-Gora | – | F | 30-45y | Local[b] | BLG | 2 | 2 | *Bovinae, Bovinae/Ovis* |
| 200 | Mamai-Gora | – | M | 21-35y | Born and lived outside of Mamai-Gora until adolescence[b] | – | 0 | 0 | – |
| 203 | Mamai-Gora | – | F | 18-21 y | Local[b] | BLG, BTN1A1, Alpha-S1-casein | 11 | 9 | Pecora, *Bovidae, Bovinae, Bovinae/Ovis,* unspecific (Mammalia) |
| 220-221 | Mamai-Gora | – | M | 40-50y | Local[b] | BLG, Alpha-S1-casein | 10 | 7 | Pecora, *Bovidae, Bovinae, Bovinae/Ovis, Capra* |

More information about the samples and recovered dietary peptides can be found in S1 and S4 Tables in S2 File, respectively. * Interpretation based on strontium isotopic data ($^{87}Sr/^{86}Sr$).

[a]Ventresca Miller et al., 2019

[b]Ventresca Miller et al., 2021

Osnyagi 3<sup>rd</sup> Field, Pereschepino, and Tsarina) and Mamai-Gora (Fig 1). Five samples (201, 206, 207, 209, and 213) were replicated from their corresponding dental calculus deposits due to an inadequate oral signature to test whether the result was caused by an initially insufficient amount of material for protein extraction. However, even with an increased amount of dental calculus in most cases (S2 Table in S2 File), the proteomes of the replicated samples still did not exhibit an adequate oral signature and were therefore not analysed further (S3 Table in S2 File).

Out of 43 samples from 28 individuals, the proteomes of only 15 samples from 11 individuals passed the oral signature assessment (Bilsk: n = 6, Mamai-Gora: n = 5) (S1 Fig in S1 File). Since the teeth had not been treated with consolidants and the extraction blanks processed alongside the samples in the dedicated clean laboratory were clean, the high failure rate was interpreted as reflecting variable preservation in the samples. Within the group of individuals who passed the oral signature authentication, a Wilcoxon rank-sum test showed a significant difference in total deamidation rates between contaminants and the group of oral cavity and dietary proteins ($p = 1.1 \times 10^{-5}$). In contrast, no significant difference was observed within the failed group ($p = 0.3$) (S2A Fig in S1 File). The deamidation rates of asparagine to aspartic acid (N2D) and glutamine to glutamic acid (Q2E), plotted from all proteomes that passed the oral signature assessment, showed higher deamidation rates for asparagine than for glutamine (S2B Fig in S1 File), consistent with the slower deamidation rate of glutamine compared to asparagine [42].

From the 11 individuals with an adequate oral signature, we recovered dietary proteins that derived from milk or dairy products in six individuals, from which three derived from Bilsk (Fig 2A) and three from Mamai-Gora (Fig 2B). Unfortunately, no other dietary proteins at distinct taxonomic levels were detected, limiting our ability to gain further insights into Scythian food systems. All of the results were correlated with excavation subsites and multi-isotopic data from previous publications, revealing insights into subsistence practices (Table 1) [7,21–23,43].

At both sites, a total of 50 peptide spectrum matches (PSMs) (Bilsk: 27, Mamai-Gora: 23) originating from the milk proteins Beta-lactoglobulin (BLG) and Beta-lactoglobulin-1 (BLG-1), Alpha-S1-casein, and the milk fat globule membrane (MFGM) protein Butyrophilin subfamily 1 member A1 (BTN1A1) were detected (S4 Table in S2 File). BLG was identified in 2 individuals from Bilsk: 143 (Tsarina, n = 9 PSMs) and 181–182 (Marchenki 9<sup>th</sup>, n = 4 PSMs), and in 3 individuals from Mamai-Gora: 192 (n = 2 PSMs), 203 (n = 5 PSMs), and 220–221 (n = 8 PSMs). In contrast, BLG-1 was identified exclusively in individual 143 from Bilsk (Tsarina, n = 7 PSMs). Alpha-S1-casein was identified in 3 individuals from Bilsk: 143 (Tsarina, n = 2 PSMs), 151–152 (Marchenki 8<sup>th</sup>, n = 2 PSMs), and 181–182 (Marchenki 9<sup>th</sup>, n = 3 PSMs), as well as in 2 individuals from Mamai-Gora: 203 (n = 4 PSMs) and 220–221 (n = 2 PSMs). While BTN1A1, was identified only in individual 203 (n = 2 PSMs) from Mamai-Gora (Fig 2).

Taxonomic assignments based on the identified milk peptide sequences included various ruminants and *Equus* (horse) (Figs 3A, 3B, and 3D), with the latter identified solely in the individual 143 from Bilsk (Tsarina) (Figs 3C and 3D). While the ruminant assignments of the identified milk peptides ranged from species-specific classifications, such as *Ovis* (sheep) and *Capra* (goat), to broader categories like the subfamilies *Caprinae* (e.g., sheep and goats) and *Bovinae* (e.g., cattle, yak, bison, water buffalo), the family *Bovidae* (e.g., cattle, sheep, and goats), and the higher infra-order Pecora (all even-toed mammals with ruminant digestion). The taxonomic grouping of "*Caprinae* and *Capra*" was specifically identified only at Tsarina (Bilsk) and Mamai-Gora, while "Pecora and *Bovidae*" and "*Bovinae* and/ or *Ovis*" were found at all sites. The latter grouping of bovines or sheep is reasoned by the aspartic acid (D) at the 6<sup>th</sup> position of the recovered BLG peptide TPEVDDEALEK which ambiguously can be identified as derived either from an unmodified bovine sequence or as a deamidated asparagine residue (N→D) of a sheep BLG peptide [44].

## Discussion

Scythian-era societies were neither homogeneously sedentary nor always mobile, instead engaging in intricate forms of social interaction, societal structures, and subsistence regimes. Along with established archaeological and stable isotopic evidence of dietary practices [7,12,14,21], this paleo-proteomic study of dental calculus supports findings of complex

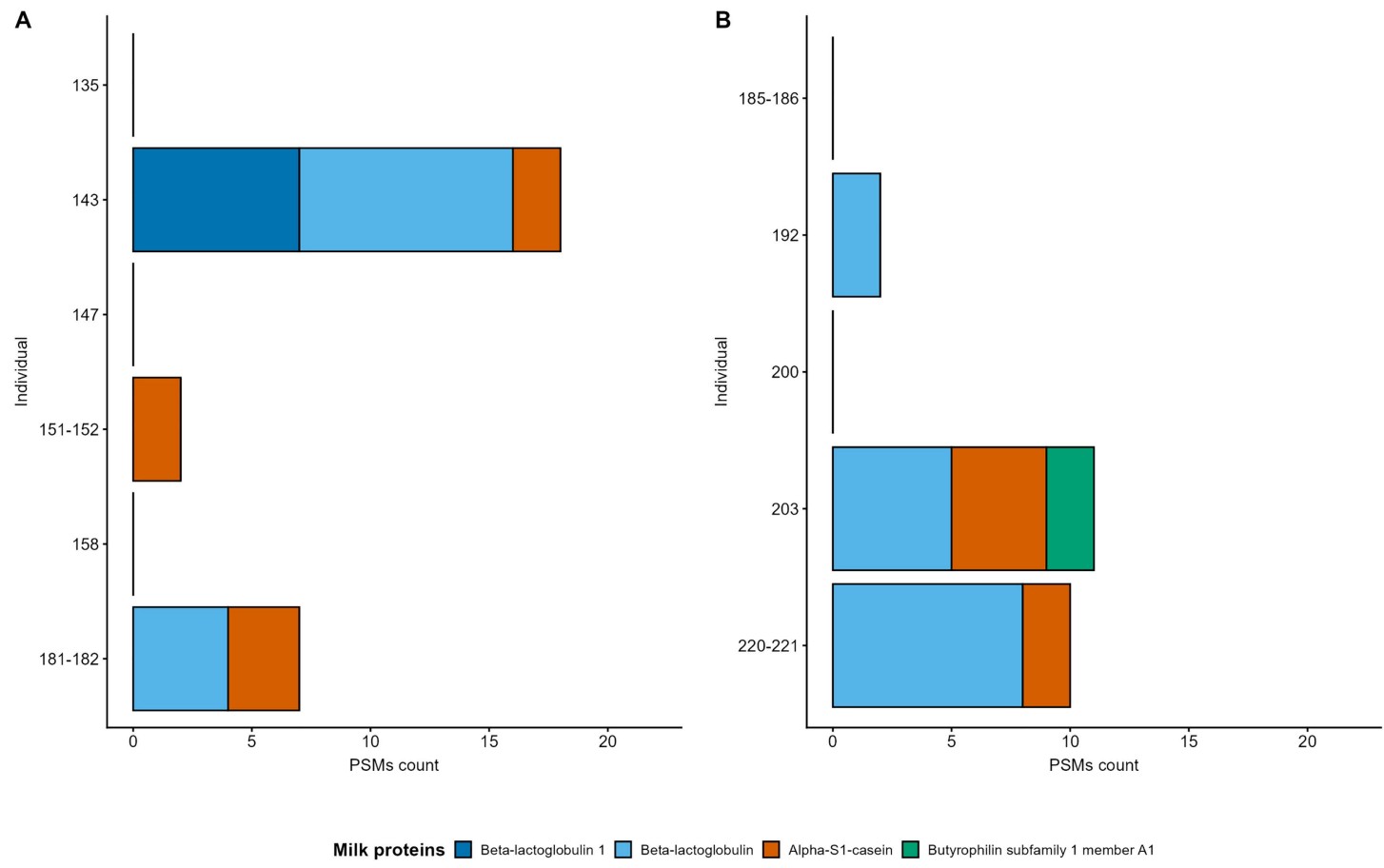

**Fig 2. Stacked bar plot of milk peptide spectrum matches (PSMs) per individual at Bilsk (A) and Mamai-Gora (B).**

subsistence economies with the identification of milk proteins derived from ruminants and horses, indicating a diverse set of agro-pastoral lifeways of Scythian populations on the Western Steppe in steppe and forest-steppe environment. However, it must be noted that an absence of certain dietary proteins in the recovered dental calculus proteomes does not mean that the corresponding individuals did not consume these foods. The incorporation, preservation, and detection of proteins in dental calculus is affected by several factors such the process and timing of calculus formation, food or beverage consistency (solid or liquid), the processing methods of dietary products, burial contexts, and the degradation of proteins over time [17,35,45]. As such, we can only comment on the presence, rather than an absence of particular taxa or tissues.

As with other steppe groups, milk was a crucial dietary resource for Scythians, as it provided a consistent, renewable, and moveable source of hydration and nutrients in the form of raw milk and processed dairy products, prior to and beyond the Iron Age [7,17,46]. Importantly, in fresh ruminant milk, casein outweighs the whey fraction (e.g., 1.1:1 in equines, 4.7:1 in bovines, and ~3.3:1 in caprines) [47,48]. This is further compounded during the processing of milk into other products where whey is removed. Overall, the ratios of casein and whey proteins found in fresh milk are not reflected in our ancient protein analysis, which is consistent with a common pattern in paleo-proteomic studies of dental calculus [17,33,34,36,40,49]. Even though the enduring survival of BLG and inverse ratio of whey and casein proteins in modern and preserved findings is still questioned [42], it can likely be explained by the complex and globular molecular structure of

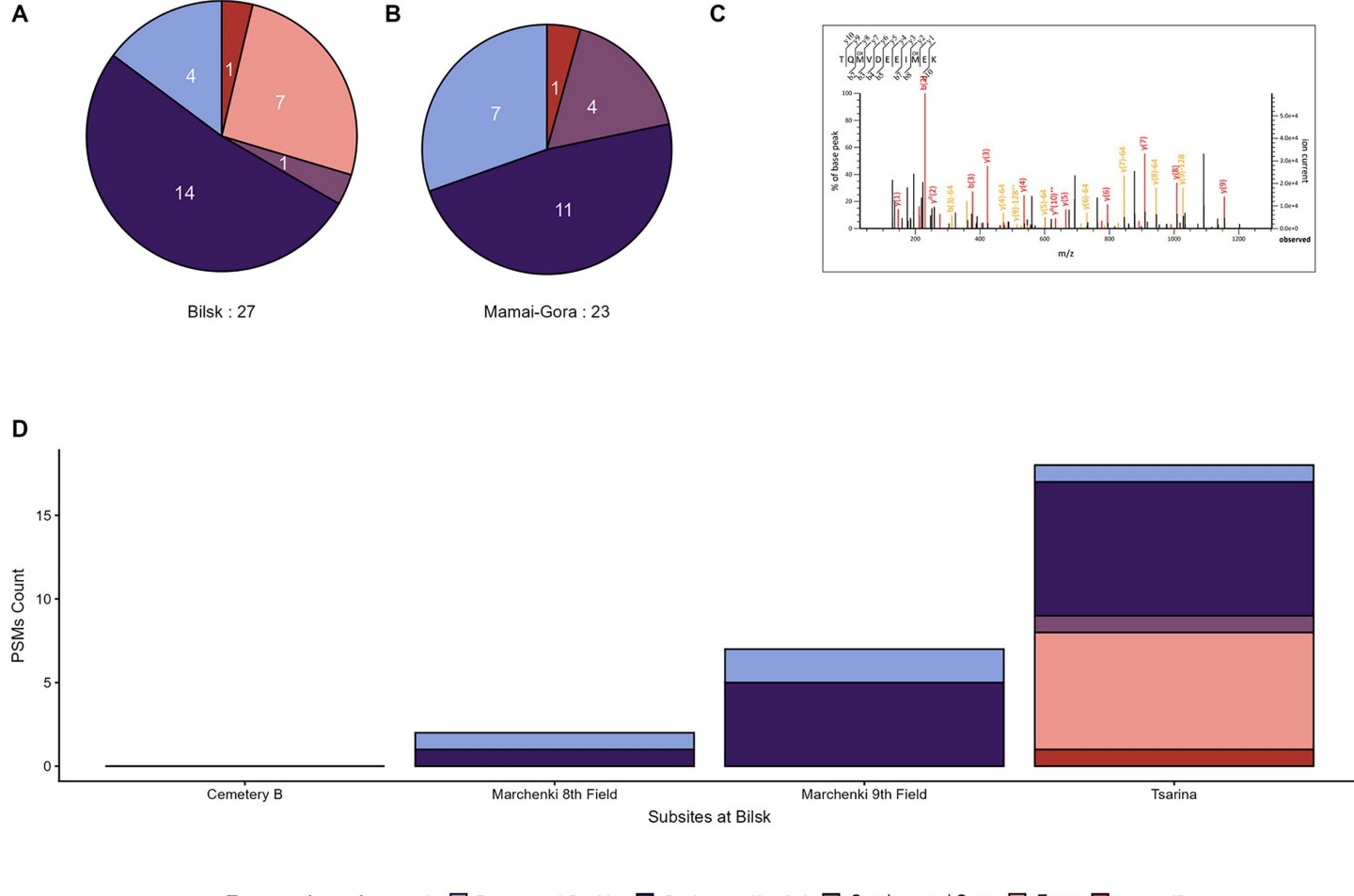

**Fig 3. Taxonomic assignments of detected peptide spectrum matches (PSMs) in individuals from the two Scythian-era sites, Bilsk and Mamai-Gora.** Pie charts and total PSMs count of taxonomic assignments at Bilsk (A) and Mamai-Gora (B). (C) depicts one MS/MS fragmentation spectra from a horse peptide recovered from an individual from Tsarina, Bilsk. (C) shows a bar plot of the taxonomic assignments of PSMs at the different subsites in Bilsk. For visualisation purposes, taxonomic assignments were summarised when the grouping did not exclude taxonomic specification, i.e., Pecora and *Bovidae*", "*Bovinae* and/ or *Ovis*", and "*Caprinae* and *Capra*".

BLG in comparison to the simpler secondary structures of casein proteins [50]. The complex and tightly bound structure of BLG presumably allows for much of the protein sequence to be less susceptible to denaturation induced by environmental factors like temperature or enzymatic attack, which properties may lead to the preservation of the protein sequences over a long period of time [45,50]. In contrast to whey and casein proteins, milk-fat-globule membrane (MFGM) proteins represent only a small fraction (~1%) of the total protein count in milk [47]. To our knowledge, we have provided the first paleo-proteomic evidence of BTN1A1 in dental calculus, as MFGM peptides have so far only been detected in archaeological material such as ceramic vessels [30], grass woven baskets [51], and preserved ancient cheese [52].

Apart from the ruminant-derived milk proteins, horse milk proteins (Beta-lactoglobulin 1) were only identified in a single individual at Bilsk (Tsarina: 143). Although the domestication and economic uses of horses have been documented before the emergence of the Scythians in the Iron Age [53,54], this presents the first direct biomolecular evidence that the Scythians included horse-derived products in their diet. Possible evidence of horse milk consumption on the Eurasian steppe

has been presented in the form of fatty acids as organic residues in ceramics from Botai in northern Kazakhstan [55] as well as, more recently, proteomic evidence found in dental calculus of western steppe dwellers from the Early Bronze Age Pontic-Caspian region [18] and those from the Late Bronze Age to the Mongol Empire (and into today) on the far eastern steppe in Mongolia [17,56].

Horses were clearly an integral part of the economic, political, and cultural life across the diverse range of Scythian populations as evidenced by findings of equestrian accoutrements, equine skeletal remains, and intricate horse-style craftworks included in burials [1,9,14,57,58]. According to historical records, the Scythians used horses not only for facilitated transportation, complex warfare, and as symbolic entities in their belief system, but also for milk and meat production. In these literary sources, the Scythians are even referred to as mare's milk drinking tribes that also consumed the alcoholic fermented horse milk beverage "*koumiss*" [14]. However, how widespread the practice was among the varied Scythian-associated communities is unclear. With the lack of evidence for horse milk consumption in most individuals from the current study, it is possible that access to *koumiss* was limited to those with cultural or political power. It could also be argued that the Scythians divided the functionality of their livestock according to the differential needs in animal husbandry, with horses serving a special purpose in society, used for riding and as pack animals, rather than for dietary purposes. While our contextualisation of scarce mare's milk consumption in the wider Scythian range of influence remains speculative, further archaeological and biomolecular data from a wider portion of Scythian-era populations can help to clarify disparities between social, cultural, and political structures.

## Conclusion

Our paleo-proteomic study of dietary foods consumed by Scythian populations provides new insights into the diverse subsistence strategies of communities in Iron Age Ukraine. Our proteomic evidence of mixed dairy pastoralism, supports previously published stable isotopic and archaeological data. Based on the recovered milk proteins, our data demonstrates that ruminant milk was consumed across the Northern Black Sea Region among Iron Age steppe populations. The scarce mare's milk protein recovery could be interpreted as Scythian-era populations were primarily focused on riding or traction-focused horse herds that were not milked, or that there were hierarchical divisions affecting access to certain foods.

For upcoming studies, we propose to increase the overall number of Iron Age study individuals from Scythian-era populations in order to conduct a more thorough proteomic analysis of individuals from both urban and more ephemeral pastoral sites across steppe and forest-steppe regions. In addition, a comparison of proteomic data between different Scythians populations, as well against those of other steppe peoples may reveal differences in diet, such as the consumption of mare's milk. An extension of additional biomolecular archaeological methods, such as ancient DNA and lipid analysis on bioarchaeological and archaeological material, but also the analysis of micro-remains in dental calculus can provide more detailed information and further clarify consistencies and differences in food systems across Scythian-era populations.

## Methods

### Sample collection and extraction

Dental calculus samples from the individuals excavated from Bilsk and Mamai-Gora were collected at the Institute of Archaeology of the National Academy of Sciences of Ukraine in Kyiv under the permission of a signed "Agreement Concerning Scientific Collaboration". The project was approved by the Director of the Institute of Archaeology of the Ukrainian Academy of Sciences. Originally, the study protocol did not include analysis of human dental calculus, but approvals were subsequently received from respective parties. Additional information regarding the ethical, cultural, and scientific considerations specific to inclusivity in global research is included in the Supporting Information (S3 File).

From the individuals with dental calculus, between 1 mg to 15 mg dental calculus powder or chunks were sampled and stored in Eppendorf tubes. Protein extractions were conducted in a dedicated, clean protein laboratory facility according

to the Single-Pot, Solid-Phase, Sample-Preparation (SP3) protocol [59], at University of Zurich (UZH). Each sample batch included one extraction blank as a negative control. Prior to measurement using high-performance liquid chromatography tandem mass spectrometry (HPLC-MS) at the Functional Genomics Centre Zurich (FGCZ), the samples were adjusted through dilution to a peptide concentration up to 0.03 µg/µl (S1 and S2 Tables in S2 File).

## Proteomic measurements

Mass spectrometry analysis was performed in the Functional Genomics Centre of Zürich (FGCZ) based on instrument availability on either the Q Exactive HF mass spectrometer (Thermo Scientific) equipped with a Digital PicoView source (New Objective) or on the Orbitrap Exploris 480 mass spectrometer (Thermo Fisher Scientific) equipped with a Nano-spray Flex Ion Source (Thermo Fisher Scientific), both coupled to an M-Class UPLC (Waters). In each instrument, solvent composition at the two channels was 0.1% formic acid for channel A and 0.1% formic acid, 99.9% acetonitrile for channel B. Column temperature was at 50°C. For each sample 2–5 µL of peptides were loaded on a commercial ACQUITY UPLC M-Class Symmetry C18 Trap Column (100Å, 5 µm, 180 µm x 20 mm, Waters) followed by ACQUITY UPLC M-Class HSS T3 Column (100Å, 1.8 µm, 75 µm X 250 mm, Waters) on the Q Exactive HF or on a commercial nanoEase MZ Symmetry C18 Trap Column (100Å, 5 µm, 180 µm x 20 mm, Waters) followed by a nanoEase MZ C18 HSS T3 Column (100Å, 1.8 µm, 75 µm x 250 mm, Waters) on the Orbitrap Exploris 480. The peptides were eluted at a flow rate of 300 nL/min. For measurement on the Q Exactive HF, after a 3 min initial hold at 5% B, a gradient from 5 to 22% B in 50 min and 22–40% B in additional 10 min was applied. For measurement on Orbitrap Exploris 480, after a 3 min initial hold at 5% B, a gradient from 5 to 22% B in 80 min and 22–32% B in additional 10 min was applied. The column was cleaned after the run by increasing to 95% B and holding 95% B for 10 min prior to re-establishing loading condition. All samples were measured in randomised order.

**Q Exactive HF.** The mass spectrometer was operated in data-dependent acquisition (DDA) mode with a maximum cycle time of 3 s, using Xcalibur (tune version: 4.4), with spray voltage set to 2.3 kV, funnel RF level at 60%, and heated capillary temperature at 275 °C. Full-scan MS spectra (350 − 1'500 m/z) were acquired at a resolution of 120'000 at 200 m/z after accumulation to an automated gain control (AGC) target value of 100'000 or for a maximum injection time of 100 ms. Precursors with an intensity above 45'000 were selected for MS/MS. Ions were isolated using a quadrupole mass filter with a 1.2 m/z isolation window and fragmented by higher-energy collisional dissociation (HCD) using a normalised collision energy of 28%. MS2 spectra were recorded at a resolution of 30'000 and a maximum injection time of 50 ms. Charge state screening was enabled, and singly, unassigned charge states and charge states higher than seven were excluded. Precursor masses previously selected for MS/MS measurement were excluded from further selection for 30 s, applying a mass tolerance of 10 ppm. The samples were acquired using internal lock mass calibration on m/z 371.1012 and 445.1200.

**Orbitrap Exploris 480.** The mass spectrometer was operated in data-dependent acquisition (DDA) mode with a maximum cycle time of 3 s, using Xcalibur (tune version: 4.4), with spray voltage set to 2.3 kV, funnel RF level at 40%, heated capillary temperature at 275 °C, and Advanced Peak Determination (APD) on. Full-scan MS spectra (350 − 1'200 m/z) were acquired at a resolution of 120'000 at 200 m/z after accumulation to a target value of 3'000'000 or for a maximum injection time of 45 ms. Precursors with an intensity above 5'000 were selected for MS/MS. Ions were isolated using a quadrupole mass filter with a 1.2 m/z isolation window and fragmented by higher-energy collisional dissociation (HCD) using a normalised collision energy of 30%. HCD spectra were acquired at a resolution of 30'000 and maximum injection time was set to Auto. The automatic gain control (AGC) was set to 100'000 ions. Charge state screening was enabled such that singly, unassigned and charge states higher than six were rejected. Precursor masses previously selected for MS/MS measurement were excluded from further selection for 20 s, and the exclusion window was set at 10 ppm. The samples were acquired using internal lock mass calibration on m/z 371.1012 and 445.1200.

The mass spectrometry proteomics data were handled using the local laboratory information management system (LIMS) [60].

## Data analysis

The raw MS/MS data files (.raw) were converted to Mascot generic files (.mgf) to be searched via in the Mascot daemon program (www.matrixscience.com, version: 2.7.0.1) [61] using the database from Wilkin and colleagues [62], which is compiled of the entirety of Swiss-Prot (downloaded: April 2021) combined with a custom dairy database used in previously published studies of steppe dairy populations [17,18,62]. The Mascot search included the following settings: carbami-domethylation of cysteine (C) as a fixed modification, and deamidation of asparagine and glutamine (N and Q) and oxidation of methionine (M) as variable modification. The instrument was set as QExactive, with precursor ions mass tolerance at 10 ppm, with allowances for one isotopic mass shift, and fragment ion mass tolerance at 0.01 Da. Trypsin was selected as the enzyme, and we allowed up to 3 missed cleavages per peptide and included peptides with charges of 2+, 3+, and 4+.

The Mascot results were filtered using the custom-made R script MS-MARGE.R (Freely available at: https://bitbucket.org/rwhagan/ms-marge/src/master/) [17,63], which resulted in three output files: a CSV file of filtered peptide spectrum matches (PSM), a FASTA file of the results, and an HTML file that contains information on false discovery rate (FDR) and filtered protein PSM counts. MS-MARGE.R contains adjustable filtering parameters for the minimum PSM count per protein and e-value cut-off (the default values used are >=2 and 0.01, respectively). Samples that failed to meet the false-discovery rate thresholds, by means protein FDR > 5% and peptide FDR > 2%, were excluded from further analysis. The false discovery rates for the samples that were further analyzed for their dietary content ranged for protein from 0–3.76% and for peptide from 0–1.16%, respectively.

To further validate all dietary PSMs with a Mascot ion score below 40, we applied the same strategy for spectral angle validation as conducted by Wilkin and colleagues in Review [64]. We compared each spectrum after removal of the precursor ion peak with the corresponding `AlphaPept_ms_generic` [65] fragment-ion prediction model using R (version 4.5.1), Bioconductor (version 3.19), and the koinar package [66,67]. For this, we used the model parameters of QE for Thermo Fischer Scientific (TFS) QExactive mass spectrometer and of 28eV for the collision energy (CE) setting. The spectral data was extracted from the TFS generated raw files using the rawrr package [68]. We considered PSMs without methionine or proline modification (Unimod:35) with spectral angle Pearson scores >= 0.70 as positively identified peptides, whereas PSMs with these modifications had to reach a threshold of >=0.60 for incorporation in the study. The latter adapted threshold is due to the limited prediction power with respect to post-translational modifications in current prediction models [64]. Based on spectral angle Pearson scores and minimum PSM count per protein (>=2), a total of six dietary PSMs had to be excluded from this study (S4 Table in S2 File).

## Assessment of oral signature for sample authentication

To assess the oral signature of the recovered dental calculus proteomes, we screened all of them for the presence of oral signature proteins, e.g., host-expressed salivary proteins and oral microbiome, and common and lab contaminants [44]. Similar as in the study of Ventresca Miller and colleagues (2023), we did not perform the database search against the Oral Signature Screening Database (OSSD) from Bleasdale, Boivin, and Richter (2021) [44,49], but rather searched within the filtered data from our searches in Mascot using an adapted version of the OSSD entries. We identified in each proteome the presence of peptide-spectrum matches (PSM) that match to oral signature proteins (oral microbiome and salivary proteins) and contaminants (common environmental and laboratory contaminants). These four databases are compiled out of previous publications [44,69,70]. For our authentication score, we divided for each proteome the oral signature PSM counts by the sum of the oral and contaminant PSM counts. These resulting scores were min-max normalized for better comparison across them. Based on the visual inspection of each proteome and the comparison of the oral and

contaminant PSM counts within the whole dataset, the threshold to pass as an authentic sample was set to 0.5. In case two proteomes from the same individual passed with an authentic oral signature, their raw MS/MS data were merged (name resulted from the combination of the single sample numbers with a hyphen) and followingly, the database search, proteomic pre-processing and oral signature assessment was repeated. Exclusively, single or merged samples that successfully passed this workflow were further analysed (S3 Table in S2 File).

In addition, a bulk deamidation analysis of asparagine and glutamine was conducted to assess possible indications of age-induced degradation [38]. For this, we compared the deamidation rates in proteins matching the laboratory and common contaminant database with those matching the oral cavity (oral microbiome and host proteins) and dietary proteins. To compare total deamidation rates between contaminants and the group of oral cavity and dietary proteins, we performed Wilcoxon rank-sum tests separately within the groups of individuals who passed and failed the oral signature authentication. Data processing, statistical testing, and visualisations were performed in R (version 4.5.1) using the packages tidyverse [71] and ggpubr [72].

### Assessment of taxonomic assignments of dietary proteins

All PSMs that were identified in the Mascot search as derived from dietary proteins were further authenticated in a Basic Local Alignment Search Tool (BLAST, edition: BLAST+ 2.12.0). In this search, the peptide sequences of interest were searched against the BLAST database of all known and hypothetical sequences in a non-redundant protein sequences (nr) database with the algorithm protein-protein BLAST (blastp). Only unique hits to taxa were assessed, whereas certain dairy peptides align to hypothetical protein sequences of bacteria species, e.g., Alpha-S1-casein peptides align with hypothetical proteins of *Jeotgalicoccus coquina* (WP_229715011), *Jeotgalicoccus aerolatus* (WP_231957245), and *Phocicoccus schoeneichii* (WP_229713947). As discussed by Wilkin and colleagues (2021), such theoretical hits are assumed to be derived from contaminations during the genome sequencing because of their high sequence similarities [18]. In addition, certain peptides from Beta-lactoglobulin also align with the lipocalin/fatty-acid binding family protein from *Staphylococcus aureus* (MBO8907834), since this predicted sequence derived from protein homology and we recovered several Beta-lactoglobulin peptides that were exclusively identified from various ruminant species, we counted these hits as Beta-lactoglobulin peptides. All figures were created in R (version 4.5.1) using the packages tidyverse [71], cowplot [73], and MetBrewer [74].

## Supporting information

**S1 File. Oral signature heatmaps of individuals from Bilsk and Mamai-Gora and bulk deamidation rates of asparagine and glutamine of proteomes that passed and failed the oral signature assessment.**
(DOCX)

**S2 File. Sample information, details on extraction and measurement method of samples, oral signature assessment, and dietary peptide spectrum matches.**
(XLSX)

**S3 File. Inclusivity in global research.**
(DOCX)

## Acknowledgments

Dr. Ludmilla Litvinova cannot be contacted as of the time of the article's publication. The corresponding author vouches for her contributions to the work as reported in the article and is unaware of potential competing interests for Dr. Ludmilla Litvinova that would have impacted or been relevant to this work. This work was supported by the University of Zurich's University Research Priority Program "Evolution in Action: From Genomes to Ecosystems" (S.W., V.J.S).

## Author contributions

**Conceptualization:** Jaruschka Pecnik, Alicia R. Ventresca Miller, Shevan Wilkin.

**Data curation:** Jaruschka Pecnik.

**Formal analysis:** Jaruschka Pecnik, Alicia R. Ventresca Miller, Christian Panse.

**Investigation:** Shevan Wilkin.

**Methodology:** Jaruschka Pecnik, Christian Panse, Laura Kunz, Antje Dittmann.

**Project administration:** Jaruschka Pecnik, Shevan Wilkin.

**Resources:** Alicia R. Ventresca Miller, James A. Johnson, Sergey Makhortykh, Ludmilla Litvinova, Svetlana Andrukh, Gennady Toschev, Michael Krützen, Verena J. Schuenemann.

**Supervision:** Shevan Wilkin.

**Validation:** Alicia R. Ventresca Miller, Christian Panse, Laura Kunz, Antje Dittmann, Shevan Wilkin.

**Visualization:** Jaruschka Pecnik.

**Writing – original draft:** Jaruschka Pecnik, Alicia R. Ventresca Miller, Shevan Wilkin.

**Writing – review & editing:** Jaruschka Pecnik, Alicia R. Ventresca Miller, Christian Panse, Laura Kunz, Antje Dittmann, James A. Johnson, Sergey Makhortykh, Ludmilla Litvinova, Svetlana Andrukh, Gennady Toschev, Michael Krützen, Verena J. Schuenemann, Shevan Wilkin.

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
