## [Decision Letter · Decision Letter 0]

5 Jun 2025

Dear Dr. Pecnik,

Thank you for submitting your manuscript to PLOS ONE. After careful consideration, we feel that it has merit but does not fully meet PLOS ONE’s publication criteria as it currently stands. Therefore, we invite you to submit a revised version of the manuscript that addresses the points raised during the review process.

**Even providing valuable information on dairy feeding habits of Scythians, the study may suffer from lack of robustness and possible insufficient number of individuals tested, as commented by reviewers. Possible risk of contamination was also mentioned as a considerable problem and the authors should meticulously address this issue. Several other methodological omissions are noted by Reviewer #2.**

We look forward to receiving your revised manuscript.

Kind regards,

Branislav T. Šiler, Ph.D.

Academic Editor

PLOS ONE

Journal Requirements:

3. In your manuscript, please provide additional information regarding the specimens used in your study. Ensure that you have reported human remain specimen numbers and complete repository information, including museum name and geographic location.

“All necessary permits were obtained for the described study, which complied with all relevant regulations.”

“No permits were required for the described study, which complied with all relevant regulations.”

For more information on PLOS ONE's requirements for paleontology and archeology research, see https://journals.plos.org/plosone/s/submission-guidelines#loc-paleontology-and-archaeology-research .

5. Please amend the manuscript submission data (via Edit Submission) to include authors J. Pecnik, A. R. Ventresca Miller, J. A. Johnson, S. Makhortykh, L. Litvinova, S. Andrukh, G. Toschev, M. Krützen, V. J. Schuenemann, and S. Wilkin.

6. Please amend your authorship list in your manuscript file to include authors Jaruschka Pecnik, Alicia Ventresca Miller, James Johnson, Sergey Makhortykh, Ludmilla Litvinova, Svetlana Andrukh, Gennady Toschev, Michael Krützen, Verena Schuenemann,  and Shevan Wilkin.

7. We note that Figure 1 in your submission contain map/satellite images which may be copyrighted. All PLOS content is published under the Creative Commons Attribution License (CC BY 4.0), which means that the manuscript, images, and Supporting Information files will be freely available online, and any third party is permitted to access, download, copy, distribute, and use these materials in any way, even commercially, with proper attribution. For these reasons, we cannot publish previously copyrighted maps or satellite images created using proprietary data, such as Google software (Google Maps, Street View, and Earth). For more information, see our copyright guidelines: http://journals.plos.org/plosone/s/licenses-and-copyright.

Reviewers' comments:

Reviewer's Responses to Questions

**Comments to the Author**

1. Is the manuscript technically sound, and do the data support the conclusions?

Reviewer #1: Yes

Reviewer #2: Partly

Reviewer #3: Yes

2. Has the statistical analysis been performed appropriately and rigorously?

Reviewer #1: N/A

Reviewer #2: No

Reviewer #3: N/A

3. Have the authors made all data underlying the findings in their manuscript fully available?

Reviewer #1: Yes

Reviewer #2: No

Reviewer #3: Yes

4. Is the manuscript presented in an intelligible fashion and written in standard English?

Reviewer #1: Yes

Reviewer #2: Yes

Reviewer #3: Yes

Reviewer #1: This paper presents original research on palaeo-proteomic analysis of human dental calculus to assess reliance on dairy products. Previous research presented in the paper have provided a broad picture of subsistence practices among the Scythian culture and the use of palaeo-proteomic analysis to assess specific contents of the diet is particularly useful as a complementary element of research.

The study provides evidence for the first palaeo-proteomic evidence of BTNS1A1 from dental calculus, a nice result that will encourage additional research by demonstrating additional potential questions.

While the sample size is small, this is the result of assessing the reliability of samples through oral signature assessment. It is more desirable for researchers to present small, pilot samples of solid data rather than inflate sample size with inadequate results. In addition, while the sample size is small, the proteomic data is well integrated with previous isotope and ceramic residue data to strengthen interpretations. The interpretation and conclusions presented in the paper are not overstated. The suggestion of limited access to horse milk consumption is presented as a possibility with a specific call for further assessment. The final paragraph of the conclusion focuses specifically on additional methods to further expand the initial patterns presented here.

I have only one correction - line 185, The sentence should start "As such"

Reviewer #2: This study analyzes 43 dental calculus samples from Scythian individuals and successfully identifies milk proteins in six individuals, including horse-derived milk proteins in one case. The results are archaeologically significant and offer important insights into subsistence strategies. However, several methodological and analytical aspects need more clarification or improvement.

1. High exclusion rate of dental calculus samples and possible contamination

Only 11 out of 43 samples passed the oral signature assessment, resulting in a very high exclusion rate. The authors should clearly address why so many samples failed to meet the criteria. For example, the following possibilities are candidates:

・Poor preservation of dental calculus

・Effects of post-excavation treatment or consolidation (e.g., glue from adhesives or stabilizers)

・Differences in burial environments or soil conditions at the archaeological sites

At present, it is unclear whether the low success rate reflects limitations of the proteomic protocol or issues related to sample integrity or handling.

2. Incomplete adherence to established reliability criteria in paleoproteomics

While the study reports the use of thresholds such as PSM ≥ 2, protein-level FDR < 5%, and peptide-level FDR < 2%, it does not employ stricter commonly used standards such as unique peptides ≥ 2 and FDR < 1%.

For example, the following publication supports the use of "unique peptide ≥ 2" to reduce false positives:

Peng et al., 2003, Journal of Proteome Research

https://doi.org/10.1021/pr025556v

Given the implications for reproducibility and interpretive confidence, the authors should explain why these stringent thresholds were not used, and what additional steps (if any) were taken to control the FDR. If such rationale cannot be sufficiently justified, the authors should consider reanalyzing the dataset using these established criteria.

3. Omission of deamidation analysis

Deamidation is one of the most commonly used indicators of authenticity in ancient protein studies. The authors should calculate the deamidation rates and show them in the manuscript. The authors mentioned Ramsøe et al. (2021) and other papers to justify the exclusion of deamidation analysis, bbut the previous study does not argue against calculating the deamidation rate. Instead, it highlights variability of deamidation rate of dental calculus across individuals and samples, and suggests that deamidation should be used in combination with multiple lines of evidence to authenticate ancient protein data. Given the high exclusion rate of calculus samples in this study, it would be especially valuable to report deamidation rates to support the authenticity of the peptide identifications.

4. Minor comments

・Integration with ancient genomics: The authors should discuss ancient genome of Scythians, particularly lactase persistence-related variants. Such genetic data could help to consider about the dietary interpretations.

・Equine protein spectra: Since the identification of equine milk consumption is discussed in detail, it would strengthen the study to show the MS2 spectra of the relevant peptides, particularly those assigned to Equus.

Reviewer #3: This manuscript presents a relevant study on an important topic: the history of dairying. The authors apply a relatively recent analytical approach to examine dental calculus contents and successfully retrieve dietary information of Scythian populations in present-day Ukraine, contributing to the growing body of literature about these Iron Age societies.

The analyses appear careful and methodologically rigorous, and the authors acknowledge the interpretive limitations of their data. They also clearly outline future research directions that could address the gaps and hypotheses raised in the discussion of the manuscript.

Their results provide evidence of mixed dairy pastoralism in the Scythian era, supporting previous isotopic and archaeological findings. Another important contribution of this study is the identification of horse milk protein in the dental calculus of one of the analyzed individuals, marking the first direct evidence of milk consumption from this species by the Scythians. This finding aligns with historical records of Scythian practices and adds a valuable layer of molecular data to our understanding of their dietary habits.

Furthermore, the identification of BTN1A1 (a MFGM peptide) in dental calculus represents a novel contribution to paleo-proteomics, as this marker had previously only been detected in archaeological artifacts (ceramic vessels and woven baskets) and ancient preserved cheese. This adds methodological value to the study and could influence future research in the field.

Limitations and Suggestions:

Despite the merits outlined above, the study is limited by its small sample size. Only two archaeological sites are represented, and out of 43 calculus samples collected, only samples from 11 individuals yielded proteomes that passed the criteria for dietary analysis in the “assessment of oral signature for sample authentication”. Moreover, milk protein was detected in only 6 individuals (3 per site), and horse milk was identified in just one case.

As the authors correctly acknowledge, generalizations based on such a small dataset should be cautiously approached. While the challenges of working with bioarchaeological material are well-known, and limited sample sizes are not uncommon, the study’s interpretive reach should be kept proportionate to the available evidence.

Based on the scarcity of horse milk protein in the samples, the authors suggest that access to horse milk may have varied within Scythian society, potentially reflecting hierarchical divisions. While this is a plausible hypothesis, it remains speculative given the data. Moreover, the absence of a dietary marker (such as horse milk protein) does not necessarily imply non-consumption, due to the fact that several factors affect calculus formation and the preservation of exogenous remains in its matrix.

The authors are commendably transparent about these caveats and argue that “further archaeological and biomolecular data from a wider portion of Scythian-era populations” could complement the investigations and fill the gaps left. Adding to this discussion, I would also suggest incorporating, whenever feasible, a multi-proxy approach in future analyses of dental calculus, combining, for example, protein, aDNA, and plant and other microremains data from samples (or subsamples) of the same individuals and contexts. This could strengthen interpretations and allow for a broader reconstruction of ancient diets.

In addition, I think it would be beneficial to include other bioarchaeological or bioanthropological data (if available) from the analyzed individuals, particularly the one with evidence of horse milk consumption. Such data could help assess whether status-related differences in diet are supported by other lines of evidence.

Specific observations:

• The authors state that “no other dietary proteins at distinct taxonomic levels were detected” but do not clarify whether this results from methodological limitations, database constraints, or a genuine absence of other dietary proteins. A brief explanation would be helpful.

• The section titled “Background and Workflow” is more appropriately categorized as a methodological description rather than a result. I suggest merging it with the “Methods” section for consistency and clarity.

• Line 118: The term “replication”, referring to five calculus samples, is unclear. Does this mean multiple samples from the same individual, or sub-sampling of a single calculus deposit? If the former, the authors should explain why additional samples were taken. Since the techniques used to access the dental calculus contents are typically destructive, it is important to note whether efforts were made to conserve material for future analysis and to document (by photographing and recording macroscopic aspects) the calculus deposits before its detachment. This is particularly relevant given the growing value of calculus as a multi-proxy resource.

Text revision:

Finally, the manuscript is clear and generally well written. However, it would benefit from some revisions and careful proofreading. Below are some issues I have identified in both the main text and the “Supplementary Information”:

Main Text:

• Line 68: “...easily accessible to water sources…” → Remove “to”: “easily accessible water sources.”

• Line 84: Replace “microfossil” with “micro-remains,” which is now the preferred and more accurate term in the context of dental calculus studies.

• Lines 87–88: The publication year for Soncin et al. is repeated. Please, revise for accuracy.

• Line 93: The map (Figure 1) should be cited earlier in the text, ideally when the archaeological sites are first introduced. I suggest replacing “see Supplementary Information” with “see Figure 1.”

• Lines 145–150: The sentence beginning with “While the ruminant assignments…” is unclear and needs rephrasing.

• Page 16: The URL provided for Geber et al. (2019) appears to be incorrect. Please verify and correct the citation.

Supplementary Information:

The section titled “Scythian settlements and cemeteries” would be better integrated into the manuscript’s Introduction and requires grammatical editing. For example:

- Lines 27–30: “As two important political and economic centres, the Scythian inhabitants engaged in agro-pastoralism, industrial workmanship, and trading, evidenced by historical records, (bio)archaeological findings, and multi-isotope analyses.” → This sentence should be restructured for clarity.

- Lines 34 & 52: Revise to “Bel’sk is located in northeastern Ukraine” and “in south-central Ukraine,” respectively.

- Line 66: In Supplementary Figure 2, “represent” should be corrected to “represents.”

**Do you want your identity to be public for this peer review?** For information about this choice, including consent withdrawal, please see our Privacy Policy

Reviewer #1: No

Reviewer #2: No

Reviewer #3: No

---

## [Author Response · Author response to Decision Letter 1]

30 Sep 2025

Response to reviewers

The authors would like to express great gratitude to the reviewers for their thoughtful reviews, constructive comments, and detailed suggestions. We greatly appreciate the opportunity to submit a major revision of our manuscript titled “Paleo-proteomic analysis of Iron Age dental calculus provides direct evidence of Scythian reliance on ruminant dairy” for your renewed consideration for submission in PLOS One. We have carefully addressed all comments and revised the manuscript accordingly. Page and line numbers refer to the revised manuscript unless otherwise noted. We also want to inform, that we adapted the site name “Bel’sk” to the Ukrainian name “Bilsk”.

Reviewers comments

Reviewer #1:

This paper presents original research on palaeo-proteomic analysis of human dental calculus to assess reliance on dairy products. Previous research presented in the paper have provided a broad picture of subsistence practices among the Scythian culture and the use of palaeo-proteomic analysis to assess specific contents of the diet is particularly useful as a complementary element of research.

The study provides evidence for the first palaeo-proteomic evidence of BTNS1A1 from dental calculus, a nice result that will encourage additional research by demonstrating additional potential questions.

While the sample size is small, this is the result of assessing the reliability of samples through oral signature assessment. It is more desirable for researchers to present small, pilot samples of solid data rather than inflate sample size with inadequate results. In addition, while the sample size is small, the proteomic data is well integrated with previous isotope and ceramic residue data to strengthen interpretations. The interpretation and conclusions presented in the paper are not overstated. The suggestion of limited access to horse milk consumption is presented as a possibility with a specific call for further assessment. The final paragraph of the conclusion focuses specifically on additional methods to further expand the initial patterns presented here.

I have only one correction - line 185, The sentence should start "As such"

Our response:

We thank the reviewer for their kind assessment of our study. We have amended the text.

Reviewer #2:

This study analyzes 43 dental calculus samples from Scythian individuals and successfully identifies milk proteins in six individuals, including horse-derived milk proteins in one case. The results are archaeologically significant and offer important insights into subsistence strategies. However, several methodological and analytical aspects need more clarification or improvement.

1. High exclusion rate of dental calculus samples and possible contamination

Only 11 out of 43 samples passed the oral signature assessment, resulting in a very high exclusion rate. The authors should clearly address why so many samples failed to meet the criteria. For example, the following possibilities are candidates:

・Poor preservation of dental calculus

・Effects of post-excavation treatment or consolidation (e.g., glue from adhesives or stabilizers)

・Differences in burial environments or soil conditions at the archaeological sites

At present, it is unclear whether the low success rate reflects limitations of the proteomic protocol or issues related to sample integrity or handling.

Our response:

We were also disappointed by the small number of samples that passed the oral signature assessment threshold, however, this outcome is not uncommon. Poor preservation and differences in burial environments / soil conditions are the most likely explanations, as they are linked, with preservation being highly dependent on factors such as microbial communities, soil conditions, and waterlogging. Preservation also varies over time and space, with some areas having amazing preservation, such as permafrost Mongolian environments. However, there have been other regions (Switzerland, Germany) where preservation may be expected to be high, but is rather the opposite. As several of these studies resulted in a lack of dietary proteins the data are often not published. While publishing negative results is ideal, these reports cannot interpret missing data and therefore generally do not meet journal requirements for publication. The preservation in this paper is not as consistently good as others, but nevertheless does contain the first proteomic dietary data for Scythians, and we believe it deserves to be published. Ideally, more samples will be assessed in the future, lending additional and more nuanced insights into their diets across a greater area of their realm of influence. There were no glues or other curatorial issues with these samples, and they were extracted using a well-established protocol for ancient human dental calculus.

However, it is important that we point out that out of 43 dental calculus samples that derived from a total of 28 individuals, 15 instead of 11 samples passed the oral signature assessment. These 15 samples with an authentic oral signature derived from 11 individuals.

Our analysis process for the assessment of the oral signature for sample authentication can be found in the method section, which explains that the proteome of all samples were first analyzed for its oral signature and only the proteomes that passed were further analyzed for the presence of dietary proteins. However, in case multiple proteomes from the same individual passed the assessment, the raw MS/MS data of these samples were merged (i.e., 151-152, 181-182, 185-186, and 220-221) and subsequently, the data pre-processing and assessment of oral signature was performed. With this approach, we wanted to make sure that we only analyze or merge data with a good oral signature.

We have adapted this sentence in the results section to make it clearer:

Out of 43 samples from 28 individuals, the proteomes of only 15 samples from 11 individuals passed the oral signature assessment (Bilsk: n=6, Mamai-Gora: n=5).

2. Incomplete adherence to established reliability criteria in paleoproteomics

While the study reports the use of thresholds such as PSM ≥ 2, protein-level FDR < 5%, and peptide-level FDR < 2%, it does not employ stricter commonly used standards such as unique peptides ≥ 2 and FDR < 1%.

For example, the following publication supports the use of "unique peptide ≥ 2" to reduce false positives:

Peng et al., 2003, Journal of Proteome Research

https://doi.org/10.1021/pr025556v

Given the implications for reproducibility and interpretive confidence, the authors should explain why these stringent thresholds were not used, and what additional steps (if any) were taken to control the FDR. If such rationale cannot be sufficiently justified, the authors should consider reanalyzing the dataset using these established criteria.

Our response:

While a 1% FDR is used widely in single organism/tissue studies of modern proteomics, we are looking at an extremely diverse and ancient metaproteome. It is generally accepted in these cases, especially dental calculus, to aim for 5% protein FDR and 2% peptide FDR. These percentages are used as initial aims, but the actual FDR percentages are always lower, and are reported in the supplementary tables. Please note that when looking at the actual peptide FDR for each sample that passed the preservation assessment in the Supplementary Tables (ST4), the rates are 0.43; 0; 0.98; 1.16; 0.22; 0.7; 0.83; 0.23; 1.03; 0; 0.62, with an average peptide FDR of 0.56.

Furthermore, as clearly stated in the data analysis paragraph of our methods section, before the FDR was calculated the data from each sample were filtered to include only proteins supported by 2 PSMs, and PSMs with an expect value lower than 0.01, which excludes unreliable PSMs.

These papers have also used an aim of 5% protein and 2% peptide cutoffs, which includes Nature, PNAS, and this journal:

Wilkin et al., 2021, Nature, https://doi.org/10.1038/s41586-021-03798-4

Wilkin et al., 2020, Nature Ecol Evol, https://doi.org/10.1038/s41559-020-1120-y

Ventresca Miller et al., 2022, PLOS One, https://doi.org/10.1371/journal.pone.0265775

Hendy et al., 2018, Nat Commun, https://doi.org/10.1038/s41467-018-06335-6

Jeong et al., 2018, PNAS, https://doi.org/10.1073/pnas.181360811

Also, we conducted an additional validation of the PSMs with Mascot ion scores below 40 using spectral angle validation. With this newly established strategy in paleo-proteomics by Wilkin and colleagues (in review), we had to exclude a total of six dietary PSMs which did not reach the Spectral angle Pearson scores or after this assessment, the data did not meet the >= 2 PSMs per protein requirement. Hence, the figures, tables, and data in the text had to be adapted accordingly. A detailed description of the analysis can be found in the method section of the manuscript and the corresponding data can be found in the S4 Table of the supplementary information.

The new paragraph in the method section of “Data analysis” reads like this:

To further validate all dietary PSMs with a Mascot ion score below 40, we applied the same strategy for spectral angle validation as conducted by Wilkin and colleagues in Review (64). We compared each spectrum after removal of the precursor ion peak with the corresponding `AlphaPept_ms_generic` (65) fragment-ion prediction model using R (version 4.5.1), Bioconductor (version 3.19), and the koinar package (66,67). For this, we used the model parameters of QE for Thermo Fischer Scientific (TFS) QExactive mass spectrometer and of 28eV for the collision energy (CE) setting. The spectral data was extracted from the TFS generated raw files using the rawrr package (68). We considered PSMs without methionine or proline modification (Unimod:35) with spectral angle Pearson scores >= 0.70 as positively identified peptides, whereas PSMs with these modifications had to reach a threshold of >=0.60 for incorporation in the study. The latter adapted threshold is due to the limited prediction power with respect to post-translational modifications in current prediction models (64). Based on spectral angle Pearson scores and minimum PSM count per protein (>=2), a total of six dietary PSMs had to be excluded from this study (S4 Table).

3. Omission of deamidation analysis

Deamidation is one of the most commonly used indicators of authenticity in ancient protein studies. The authors should calculate the deamidation rates and show them in the manuscript. The authors mentioned Ramsøe et al. (2021) and other papers to justify the exclusion of deamidation analysis, bbut the previous study does not argue against calculating the deamidation rate. Instead, it highlights variability of deamidation rate of dental calculus across individuals and samples, and suggests that deamidation should be used in combination with multiple lines of evidence to authenticate ancient protein data. Given the high exclusion rate of calculus samples in this study, it would be especially valuable to report deamidation rates to support the authenticity of the peptide identifications.

Our response:

We agree with the reviewer and we have now included a bulk deamidation assessment for this study. Details about the method are described in the “Assessment of oral signature for sample authentication” section and results of the assessment can be found in the results section such as the corresponding figure in the supplementary information (S2 Fig).

The new section in the methods reads now like this:

In addition, a bulk deamidation analysis of asparagine and glutamine was conducted to assess possible indications of age-induced degradation (38). For this, we compared the deamidation rates in proteins matching the laboratory and common contaminant database with those matching the oral cavity (oral microbiome and host proteins) and dietary proteins. To compare total deamidation rates between contaminants and the group of oral cavity and dietary proteins, we performed Wilcoxon rank-sum tests separately within the groups of individuals who passed and failed the oral signature authentication. Data processing, statistical testing, and visualisations were performed in R (version 4.5.1) using the packages tidyverse (71) and ggpubr (72).

The new paragraph in the results section reads like this:

Out of 43 samples from 28 individuals, the proteomes of only 15 samples from 11 individuals passed the oral signature assessment (Bilsk: n=6, Mamai-Gora: n=5) (S1 Fig). Within the group of individuals who passed the oral signature authentication, a Wilcoxon rank-sum test showed a significant difference in total deamidation rates between contaminants and the group of oral cavity and dietary proteins (p = 1.1 × 10⁻⁵). In contrast, no significant difference was observed within the failed group (p = 0.3) (S2A Fig). The deamidation rates of asparagine to aspartic acid (N2D) and glutamine to glutamic acid (Q2E), plotted from all proteomes that passed the oral signature assessment, showed higher deamidation rates for asparagine than for glutamine (S2B Fig), consistent with the slower deamidation rate of glutamine compared to asparagine (42).

4. Minor comments

・Integration with ancient genomics: The authors should discuss ancient genome of Scythians, particularly lactase persistence-related variants. Such genetic data could help to consider about the dietary interpretations.

Our response:

LP allele frequency in the Iron Age is very low, around 2-3% in Scythians, which is the same as all other Europeans and Central Asians at the time. LP was low until much later in time, even though people were clearly consuming dairy for over 5000 years. To understand the nuances regarding the link between LP allele and Scythian’s ability to drink fresh milk we will need more information past gut microbiomes and pre-consumptions methods that reduce or break down lactose. Interestingly, today in Mongolia (as occurred over the last 5,000 years) people consume large amounts of fresh milk, especially in Spring-Autumn, without a genetic adaptation to break down the lactose into glucose and galactose so that it can be absorbed. Therefore, whether the Scythians did or have the allele may not have affected their ability to drink milk without negative side effects. As we have very little information on milk consumption over the entirety of the Scythian range of influence, we would like to reserve this conversation for when we have more data.

Unterländer et al., 2017, Nat Commun, https://doi.org/10.1038/ncomms14615

Evershed et al., 2022, Nature, https://doi.org/10.1038/s41586-022-05010-7

Segurel et al., 2020, PLoS Biol, https://doi.org/10.1371/journal.pbio.3000742

・Equine protein spectra: Since the identification of equine milk consumption is discussed in detail, it would strengthen the study to show the MS2 spectra of the relevant peptides, particularly those assigned to Equus.

Our response

We have now added an MS/MS fragmentation spectra of a horse peptide to the newly created Figure 3.

Reviewer #3:

This manuscript presents a relevant study on an important topic: the history of dairying. The authors apply a relatively recent analytical approach to examine dental calculus contents and successfully retrieve dietary information of Scythian populations in present-day Ukraine, contributing to the growing body of literature about these Iron Age societies.

The analyses appear careful and methodologically rigorous, and the authors acknowledge the interpretive limitations of their data. They also clearly outline future research directions that could address the gaps and hypotheses raised in the discussion of the manuscript.

Their results provide evidence of mixed dairy pastoralism in the Scythian era, supporting previous isotopic and archaeological findings. Another important contribution of this study is the identification of horse milk protein in the dental calculus of one of the analyzed individuals, marking the first direct evidence of milk consumption from this species by the Scythians. This finding aligns with historical records of Scythian practices and adds a valuable layer of molecular data to our understanding of their dietary habits.

Furthermore, the identification of

---

## [Decision Letter · Decision Letter 1]

26 Nov 2025

Dear Dr. Pecnik,

Thank you for submitting your manuscript to PLOS ONE. After careful consideration, we feel that it has merit but does not fully meet PLOS ONE’s publication criteria as it currently stands. Therefore, we invite you to submit a revised version of the manuscript that addresses the points raised during the review process.

**Several additional clarifications should be made, according to Reviewer #4.**

We look forward to receiving your revised manuscript.

Kind regards,

Branislav T. Šiler, Ph.D.

Academic Editor

PLOS ONE

Journal Requirements:

Reviewers' comments:

Reviewer's Responses to Questions

**Comments to the Author**

Reviewer #2: All comments have been addressed

Reviewer #4: (No Response)

2. Is the manuscript technically sound, and do the data support the conclusions?

Reviewer #2: Partly

Reviewer #4: Yes

3. Has the statistical analysis been performed appropriately and rigorously?

Reviewer #2: N/A

Reviewer #4: Yes

4. Have the authors made all data underlying the findings in their manuscript fully available?

Reviewer #2: Yes

Reviewer #4: Yes

5. Is the manuscript presented in an intelligible fashion and written in standard English?

Reviewer #2: Yes

Reviewer #4: Yes

Reviewer #2: (No Response)

Reviewer #4: I enjoyed reading and assessing this manuscript on the palaeoproteomics of Scythian dental calculus. The topic addressed is important for understanding the occurrence and diversity of dairy use in the Iron Age steppe context. The study is clearly written and appears to be methodologically careful, particularly following the recent manuscript revisions. The identification of BTN1A1 is an interesting and valuable aspect of the work. The reports of horse milk assignment are quite well supported. Overall, this is a well-conducted and interesting contribution, and I have only a few comments that should be straightforward to address.

Sample size

The sample size is small but carefully filtered, which is entirely appropriate. This approach is preferable to inflating n with data of uncertain or poor quality, and it reflects a commendable level of caution in data selection.

Clarification from the response to Reviewer 2

In the response to Reviewer 2, the authors state that “the high failure rate is interpreted as reflecting variable preservation rather than laboratory contamination, as no consolidants were used and blanks were clear.” This is a reasonable explanation, but it is not clear whether this statement appears explicitly in the main text. If not, it would be good to include a brief mention of this rationale in the manuscript itself for transparency.

Title and framing

The title emphasises milk as a staple through the word “reliance.” While this is an engaging framing, it might be a little strong given that the dataset comprises only six samples with milk proteins from two sites. This represents solid direct evidence for dairy consumption, and likely regular consumption, but “reliance” may somewhat overstate the strength of the direct data unless the authors’ argument is that these proteomic findings confirm or complement historical accounts of horse milk use among Scythian groups. Some slight moderation or clarification of this framing could be considered.

Threshold at 0.5

The choice of a pass threshold at 0.5 “based on inspection” could benefit from a brief justification in the Methods section. For instance, was this threshold selected because it effectively distinguishes between highly contaminated samples and oral-rich proteomes? A short explanatory sentence would help readers understand how this decision was made and improve methodological clarity.

Deamidation

The inclusion of deamidation assessment in the revised manuscript is reassuring and significantly strengthens confidence in the authenticity of the results. This is a valuable addition.

Protein naming consistency

There appear to be inconsistencies in the naming of BTN1A1 / BTNS1A1 / BTS1A1. These should be standardised throughout the manuscript for clarity and accuracy.

Data availability

In the tracked version, the following statement appears:

“Mass spectrometry proteomics data have been deposited on MassIVE (massive.ucsd.edu; MSV000092635). For reviewer access use the username: MSV000092635_reviewer and the password: Scythian-Paper!.”

This is perfectly acceptable during peer review but would raise a minor concern if carried into the published version, as it suggests the repository may not yet be fully public. PLOS will expect the dataset to be completely open (no login required) upon publication.

In the final version, the Data Availability statement should therefore read simply:

“Mass spectrometry proteomics data are available on MassIVE (MSV000092635).”

No usernames or passwords should be included.

Table 1 – Locality of individual (Sr)

In Table 1, the column “Locality of individual (Sr)” includes the entry “born outside and moved as a child.” I understand this derives from a previous study, but it remains an interpretation rather than a directly observed result. While it may be a strong inference, it is still an inference. The column heading or note should make clear that this is an interpretive statement rather than a direct measurement.

Overall, this is a thoughtful and well-constructed study that makes a meaningful contribution to the growing field of ancient proteomics. With a few minor clarifications and adjustments as suggested above, it will make a solid addition to the literature.

**Do you want your identity to be public for this peer review?** For information about this choice, including consent withdrawal, please see our Privacy Policy

Reviewer #2: No

Reviewer #4: **Yes: ** Robert C. Power

---

## [Author Response · Author response to Decision Letter 2]

5 Dec 2025

Response to reviewers

We would like to express our sincere gratitude to the reviewers for their careful evaluation and valuable insights. We are very thankful that we have the opportunity to submit the following minor revisions of our manuscript titled “Paleo-proteomic analysis of Iron Age dental calculus provides direct evidence of Scythian reliance on ruminant dairy” for the consideration for submission in PLOS One. We have carefully addressed all comments and revised the manuscript accordingly. Page and line numbers refer to the revised manuscript unless otherwise noted.

Reviewers comments

Reviewer #4:

I enjoyed reading and assessing this manuscript on the palaeoproteomics of Scythian dental calculus. The topic addressed is important for understanding the occurrence and diversity of dairy use in the Iron Age steppe context. The study is clearly written and appears to be methodologically careful, particularly following the recent manuscript revisions. The identification of BTN1A1 is an interesting and valuable aspect of the work. The reports of horse milk assignment are quite well supported. Overall, this is a well-conducted and interesting contribution, and I have only a few comments that should be straightforward to address.

Sample size

The sample size is small but carefully filtered, which is entirely appropriate. This approach is preferable to inflating n with data of uncertain or poor quality, and it reflects a commendable level of caution in data selection.

Clarification from the response to Reviewer 2

In the response to Reviewer 2, the authors state that “the high failure rate is interpreted as reflecting variable preservation rather than laboratory contamination, as no consolidants were used and blanks were clear.” This is a reasonable explanation, but it is not clear whether this statement appears explicitly in the main text. If not, it would be good to include a brief mention of this rationale in the manuscript itself for transparency.

Our response:

Thank you very much for your thoughtful review and kind words. We agree that the manuscript could benefit from our explanation to reviewer #2 from our first revision, thus we have added the following sentence to the results section from line 128 to 131: Since the teeth had not been treated with consolidants and the extraction blanks processed alongside the samples in the dedicated clean laboratory were clean, the high failure rate was interpreted as reflecting variable preservation in the samples.

Title and framing

The title emphasises milk as a staple through the word “reliance.” While this is an engaging framing, it might be a little strong given that the dataset comprises only six samples with milk proteins from two sites. This represents solid direct evidence for dairy consumption, and likely regular consumption, but “reliance” may somewhat overstate the strength of the direct data unless the authors’ argument is that these proteomic findings confirm or complement historical accounts of horse milk use among Scythian groups. Some slight moderation or clarification of this framing could be considered.

Our response:

Thank you very much for your insights. We believe that the title, “Paleo-proteomic analysis of Iron Age dental calculus provides direct evidence of Scythian reliance on ruminant dairy,” accurately reflects the findings of our manuscript, which in our study are interpreted in the context of existing archaeological, historical, and multi-isotopic evidence. Our results show that preserved dietary peptides provide direct evidence for the consumption of ruminant dairy among multiple individuals at both studied sites, whereas horse milk was detected in only one individual. Therefore, the term “reliance” reflects the integration of these proteomic findings with the broader archaeological, historical, and stable isotopic evidence, indicating that ruminant dairy was likely a regular and significant component of their diet. In contrast, the limited horse milk peptide findings do not allow us to make the same claim.

Threshold at 0.5

The choice of a pass threshold at 0.5 “based on inspection” could benefit from a brief justification in the Methods section. For instance, was this threshold selected because it effectively distinguishes between highly contaminated samples and oral-rich proteomes? A short explanatory sentence would help readers understand how this decision was made and improve methodological clarity.

Our response:

Thank you very much for pointing this out. We have adapted to text for further clarification to the following (lines 380 – 381): Based on the visual inspection of each proteome and the comparison of the oral and contaminant PSM counts within the whole dataset, the threshold to pass as an authentic sample was set to 0.5.

Deamidation

The inclusion of deamidation assessment in the revised manuscript is reassuring and significantly strengthens confidence in the authenticity of the results. This is a valuable addition.

Protein naming consistency

There appear to be inconsistencies in the naming of BTN1A1 / BTNS1A1 / BTS1A1. These should be standardised throughout the manuscript for clarity and accuracy.

Our response:

Thank you very much for noticing the inconsistencies in the naming of Butyrophilin subfamily 1 member A1. We have adapted all of those instances to the correct name of BTN1A1 (i.e. lines: 156, 162, and 221, and Table 1).

Data availability

In the tracked version, the following statement appears:

“Mass spectrometry proteomics data have been deposited on MassIVE (massive.ucsd.edu; MSV000092635). For reviewer access use the username: MSV000092635_reviewer and the password: Scythian-Paper!.”

This is perfectly acceptable during peer review but would raise a minor concern if carried into the published version, as it suggests the repository may not yet be fully public. PLOS will expect the dataset to be completely open (no login required) upon publication.

In the final version, the Data Availability statement should therefore read simply:

“Mass spectrometry proteomics data are available on MassIVE (MSV000092635).”

No usernames or passwords should be included.

Our response:

Thank you for your comment. We have already made the data public on MassIVE during the last revision process and therefore the sentence with the password credentials was flagged as deleted in the manuscript with track changes document. Nevertheless, we have amended the text according to your suggestion, so it reads clearer.

Table 1 – Locality of individual (Sr)

In Table 1, the column “Locality of individual (Sr)” includes the entry “born outside and moved as a child.” I understand this derives from a previous study, but it remains an interpretation rather than a directly observed result. While it may be a strong inference, it is still an inference. The column heading or note should make clear that this is an interpretive statement rather than a direct measurement.

Our response:

Thank you for your comment. We agree that the column could benefit from clarification. Therefore, we have amended the column title with an asterisk and added an explanation at the bottom of the table (line 149), which reads like this: * Interpretation based on strontium isotopic data (87Sr/86Sr).

Overall, this is a thoughtful and well-constructed study that makes a meaningful contribution to the growing field of ancient proteomics. With a few minor clarifications and adjustments as suggested above, it will make a solid addition to the literature.

Our response:

We want to express at this point again great gratitude to the careful review and thoughtful comments of Robert C. Power.

---

## [Editor Report · Decision Letter 2]

8 Dec 2025

Paleo-proteomic analysis of Iron Age dental calculus provides direct evidence of Scythian reliance on ruminant dairy

PONE-D-25-05635R2

Dear Dr. Pecnik,

We’re pleased to inform you that your manuscript has been judged scientifically suitable for publication and will be formally accepted for publication once it meets all outstanding technical requirements.

Kind regards,

Branislav T. Šiler, Ph.D.

Academic Editor

PLOS One
---

## [Editor Report · Acceptance letter]

PONE-D-25-05635R2

PLOS One

Dear Dr. Pecnik,

I'm pleased to inform you that your manuscript has been deemed suitable for publication in PLOS One. Congratulations! Your manuscript is now being handed over to our production team.

Kind regards,

on behalf of

Dr. Branislav T. Šiler

Academic Editor

PLOS One